# Can the Determination of HE4 and CA125 Markers Affect the Treatment of Patients with Endometrial Cancer?

**DOI:** 10.3390/diagnostics11040626

**Published:** 2021-03-31

**Authors:** Aneta Cymbaluk-Płoska, Paula Gargulińska, Michał Bulsa, Sebastian Kwiatkowski, Anita Chudecka-Głaz, Kaja Michalczyk

**Affiliations:** 1Department of Gynecological Surgery and Gynecological Oncology of Adults and Adolescents, Pomeranian Medical University, Al. Powstańców Wielkopolskich 72, 70-111 Szczecin, Poland; p.gargulinska@wp.pl (P.G.); michal.bulsa@gmail.com (M.B.); anitagl@poczta.onet.pl (A.C.-G.); kajamichalczyk@wp.pl (K.M.); 2Department of Obstetrics and Gynecology, Pomeranian Medical University, Al. Powstańców Wielkopolskich 72, 70-111 Szczecin, Poland; kwiatkowskiseba@gmail.com

**Keywords:** endometrial cancer, recurrence, CA125, HE4, serum marker

## Abstract

The aim of our research was to determine the use of CA125 and HE4 as prognostic factors in patients with different clinical staging of endometrial cancer. Sixty-two patients with advanced endometrial cancer and 287 patients with early stage endometrial cancer participated in the study. Based on the results obtained in the study, the cut-off value for HE4 was established at 186 pmol/l and correlated with the possibility of cytoreductive surgery in patients with recurrent endometrial cancer. Univariate logistic regression revealed that serum concentrations for the median CA125 correlated with DFS (HR = 1.76, *p* = 0.033) and OS (HR = 1.42, *p* = 0.025), while the median of HE4 marker correlated with DFS (HR = 1.96, *p* = 0.015) and OS (HR = 1.83, *p* = 0.004). In the multivariate analysis, a decrease in CA125 level below normal range correlated positively with DFS and OS (HR = 1.45, *p* = 0.026; HR = 1.38, *p* = 0.037). HE4 levels correlated with DFS as follows: values below the normal range (HR = 2.31, *p* = 0.01), and with OS (HR = 1.89, *p* = 0.004). Based on the results obtained in the study, we found that HE4 is a sensitive tool for predicting the risk of recurrence and overall survival in patients with endometrial cancer.

## 1. Introduction

Endometrial cancer is one of the most frequent gynecologic cancers in developed countries. The incidence of endometrial cancer has significantly increased in Central and Eastern Europe. An estimated incidence of endometrial cancer in Central and Eastern Europe in 2018 was 26.2 per 100,000 people per year [1]. Even though about 80% of patients are recognized at an early stage and have a good prognosis and low recurrence rates [2,3], the problem of advanced cases affects about 20% of patients with endometrial cancer, and is a real challenge for oncological gynecologists. The treatment of advanced cases is mainly based on chemotherapy. In previous publications, HE4 and CA125 markers have been presented as proteins that can be used in diagnosis, and could be used as potential prognostic factors in endometrial cancer [4,5]. The abovementioned markers are commonly used to differentiate malignant adnexal lesions in clinical practice. They can also be used to monitor the neo- and adjuvant treatment of ovarian cancer patients. In particular, normalization of the HE4 marker after the completion of the first line of chemotherapy is thought to be a good prognostic factor in patients with ovarian cancer [6]. Considering that most of the patients with advanced forms of endometrial cancer have elevated HE4 and CA125 markers, and in the case of relapse, a vast majority of them show an increase in at least one of the assessed biomarkers. Therefore, it seems reasonable to test the utility of HE4 and CA125 as potential markers that could predict patients’ sensitivity to platinum-based chemotherapy, the surgical outcome, and the survival parameters. The aim of our research was to determine the usefulness of CA125 and HE4 serum markers in patients with advanced and recurrent endometrial cancers.

## 2. Materials and Methods

### 2.1. Patient Characteristics

This prospective study was conducted at the Department of Gynecological Surgery and Gynecological Oncology of Adults and Adolescents at the Pomeranian Medical University. Having read the information concerning the study, all of the patients included in the study signed a written consent to participate. Each patient was allowed to ask questions and was provided with comprehensive information. The bioethical commissions’ consent to the study was approved by the local ethical committee of the Pomeranian Medical University (Resolution number: KB-0012/77/12 of the Bioethics Committee of the Pomeranian University of Medicine in Szczecin on 13 October 2012).

All of the patients enrolled in the study underwent endometrial biopsy, abrasions, or hysteroscopy. Based on the obtained histopathological results, they were qualified for radical surgery involving the removal of the uterus with appendages and hip salping in less advanced stages. In advanced stages of endometrial cancer, radical hysterectomy and lymphadenectomy were performed. Total pelvic exenteration was performed in stage IV patients. Patient analysis was conducted depending on the hormonal status, dividing the population into two subgroups: pre- and postmenopausal patients. Patients were qualified as postmenopausal if their last menstrual period was more than 12 months ago or if their FSH level was above 30 U/L. Patients aged 36 to 79 were enrolled in the study. The mean age of the patients enrolled in the study was 60.8 years.

### 2.2. Biochemical Analysis of HE4 and CA125

At the time of hospital admittance for surgical or chemotherapeutic treatment, based on the clinical staging of the patient, a sample of five milliliters of blood was collected from each patient in order to determine HE4 and CA125 concentrations. The determination of the tested biomarkers was performed on the same day, within 2 h from the sample collection. To determine the levels of the assessed markers, postoperative patients had a follow-up examination at the outpatient clinic after surgery, and every 3 months until recurrence or the end of the study (52 months). Serum HE4 concentrations were measured using the Elecsys ECLIA (electrochemiluminescence immunoassay) assay from Roche (Basel, Switzerland), running on the cobas e 601 analyzers. The detection range of HE4 was 15–1500 pmol/L. For the purpose of the study, patients were classified into two main categories: pre-menopausal and post-menopausal, in accordance with their hormonal status. HE4 and CA125 are serum markers commonly used in the diagnosis of ovarian cancer. In our study, we considered the normal value of HE4 to be <70 pmol/L, as is established for patients with ovarian cancer. CA125 marker serum levels were determined using the ARCHITECT CA125 II assay on the ARCHITECT 2200SR system. The reference value for both premenopausal and postmenopausal patients was 35 U/mL (equal to the reference value used in ovarian cancer diagnostics).

### 2.3. Statistical Analysis

The statistical analysis was performed using Statistica version 10.0. The concentrations of biomarkers were presented for individual groups and subgroups in the form of median and ranges. Due to the lack of normal distribution, group comparisons were performed using a non-parametric *U*-Mann–Whitney test. For the selected groups, the receiver operating characteristic (ROC) curves were obtained, and the area under the curve (AUC) was calculated with 95% confidence intervals according to the nonparametric method of DeLong. In order to evaluate the influence of the HE4 and CA125 markers on DFS (time from the end of chemotherapy to relapse) and OS (time from the initial diagnosis to death), Kaplan–Meier curves and the log-rank test were used. The Cox regression model was used for univariate and multivariate analysis. The variables used in the multivariate analysis included patients’ age, staging, grading, and appropriate cut-off points for the serum concentrations of HE4 and CA125. The logistic regression model was used to assess the ability of HE4 and CA125 to predict treatment outcomes, i.e., disease recurrence, onset, or less than three years of survival. The results were considered significant for *p* < 0.05.

## 3. Results

All patients expressing their consent to participate in the study were followed-up in the clinic within the duration of the study (52 months of confirmed endometrial cancer). In the beginning, 81 patients diagnosed with advanced endometrial cancer and 300 with early-stage endometrial cancer (FIGO I or FIGO II) were enrolled in the study. Ten patients refused the proposed treatment; 3 patients died of non-oncological causes during the duration of the study; and 12 patients were excluded due to severe renal failure, elevated creatinine levels, and low GFR, which could affect the results of the HE4 marker. Moreover, one patient was excluded due to rheumatoid arthritis, one due to myasthenia gravis, and four patients due to endometriosis. One patient was also excluded due to idiopathic pulmonary fibrosis. Ultimately, 349 patients were enrolled in the study. Table 1 presents the detailed characteristics of the patients.

### 3.1. Correlation between HE4, CA125 and Prognostic Factors

In Table 2, we present the values of HE4 and CA125 obtained as a result of blood examinations. Correlations between the prognostic factors and the levels of HE4 and CA125 were evaluated.

We found that there was no statistically significant difference in serum HE4 and CA125 concentrations between the patients diagnosed with type I (endometrioid) and patients with type II endometrial cancer (non-endometrioid). Moreover, the mean CA125 concentration did not significantly differ between patients with and without lymph vessel involvement. The mean CA125 concentration in patients with lymph node invasion was 53.5 IU/mL, and in patients without the lymph node invasion, it was equal to 38.8 IU/mL (*p* = 0.058).

The serum levels of HE4 and CA125 were significantly higher in patients presenting with advanced stages of the disease (*p* = 0.001 and *p* = 0.003, respectively). Similar correlations were noticed in patients with poorly differentiated tumors (*p* = 0.001 and *p* = 0.02), lymph node metastases (*p* = 0.042 and *p* = 0.01), and deeper myometrial infiltration (*p* = 0.001 and *p* = 0.02).

### 3.2. Evaluation of HE4 and CA125 as Diagnostic Tests

In order to evaluate the diagnostic values of HE4 and CA125, ROC curves were plotted, and the areas under the ROC curves (AUC) were calculated. Based on the ROC curves obtained in the study—Figure 1.

Considering the diagnostic possibilities of the HE4 and CA125 markers, depending on the clinical staging and histopathological differentiation, we found that the histopathological grading (G1 versus G3) and FIGO staging (FIGO III and IV versus I and II) adequate areas of the curve equaled 0.78/0.66 and 0.88/0.71, respectively—Figure 2 and Figure 3.

As a part of the study, we have calculated the sensitivity of the studied markers. The sensitivity for HE4 concentration was higher in the group of premenopausal patients than in post-menopausal patients (72% vs. 68%). However, the situation was different in the case of CA125, where the sensitivity was higher among postmenopausal patients (82% vs. 78%).

The specificity of the HE4 and CA125 concentration was, respectively, 94% vs. 61% in premenopausal patients and 88% vs. 74% in postmenopausal patients, and 91% vs. 66% in the whole study group.

In case of recurrence, after the exclusion of a disseminated form of the disease, patients presenting with a single recurrent tumor were qualified for cytoreductive surgery. Based on the ROC curves obtained in the study, we established the cut-off point for HE4 at 186 pmol/L, which should allow for total debulking surgery (to R0) in patients with endometrial cancer. There was no statistically significant correlation with the possibility of cytoreduction for the CA125 marker. The differences in the concentrations of both markers were statistically significant at the time of diagnosis and of tumor recurrence—Figure 4 and Figure 5. Relapse of the disease occurred in patients according to the initial stage of the disease. A detailed breakdown depending on the staging is presented in Table 3. Table 4 also shows the median values of the HE4 and CA125 markers at the time of diagnosis and later, depending on the time of observation or relapse.

The exact distribution of the HE4 and CA125 marker’s median concentrations at the time of diagnosis, follow-up, and disease recurrence is presented in Table 5 and Table 6.

### 3.3. Survival Analysis Using the Kaplan–Meier Curves and Cox Proportional Hazard Regression

We have performed a statistical analysis using the Kaplan–Meier survival curves and log-rank tests. They revealed a statistically significant inverse correlation between the median levels of HE4 and the time of disease-free survival (*p* = 0.02). Patients presenting with values above the median were characterized by disease-free survival time that was shorter by 9.9 months.

Similarly, high serum baseline HE4 levels correlated with shorter overall survival of patients who presented with serum HE4 levels above the median and cut-off value (*p* = 0.01 and *p* = 0.003, respectively). The correlations were presented on Figure 6 and Figure 7. Additionally, high concentrations of CA125 correlated with shorter survival of patients. Serum concentrations of CA125 that were higher than the median value of the cut-off points were characterized by overall survival that was shorter by 5.8 months—*p* = 0.03.

### 3.4. Univariate and Multivariate Analysis

Using the univariate analysis models for both HE4 and CA125, we have demonstrated that age, clinical staging, and lymph node metastases had the greatest impact on disease-free survival. Additionally, the histopathological differentiation of the tumor was found to influence the level of HE4, and affect patients’ DFS. In the univariate analysis, the cut-off points for HE4 and CA125 were found to have the greatest impact on patients’ disease-free and overall survival.

In the multivariate analysis, all of the analyzed concentration points for HE4 were statistically significant; however, the cut-off point and the 95th percentile value of the initial serum HE4 had the greatest impact on DFS, and were, respectively, HR = 2.31, *p* = 0.01 and HR = 1.41, *p* = 0.04.

The cut-off point and the 95th percentile value for the serum concentration of HE4 were also found to have the greatest impact on overall survival. (HR = 1.89, *p* = 0.004 and HR = 1.62, *p* = 0.022, respectively). Using multivariate analysis, we have demonstrated that the factors selected in the study (age, staging, grading) influenced HE4 concentration and had a significant impact on the prolongation of OS (HR = 1.21, *p* = 0.003; HR = 1.18, *p* = 0.03; HR = 1.62, *p* = 0.022; HR = 1.89, *p* = 0.004; respectively), at the median, the 75th percentile of the baseline, the 95th percentile of the baseline, and the cut-off point. In the case of the CA125 marker, the effect on OS was statistically significant only for the median and the cut-off point of 35 IU/mL (HR = 1.22, *p* = 0.024; HR = 1.38, *p* = 0.037).

## 4. Discussion

Postmenopausal or perimenopausal bleeding is one of the most common causes of patients’ referral to gynecologists. As it is usually the first clinical sign of endometrial cancer, and it often allows for early diagnosis of the malignancy process. When compared to ovarian cancer, which often presents with non-specific symptoms and is considered to be a silent killer, the presence of postmenopausal or perimenopausal bleeding enables a rapid diagnosis of endometrial cancer. In our study, we showed that elevated serum levels of both HE4 and CA125 correlated with the already recognized prognostic factors of endometrial cancer, such as the clinical staging, the depth of the myometrium infiltration, and lymph node metastases. As it is commonly known, the specificity of HE4 is higher than of CA125. In non-oncological cases, it was shown to increase in patients suffering from renal failure, as well as in patients undergoing psychiatric treatment with neuroleptics. We did not include such patients in our study. CA125 is a marker whose levels can be altered by ongoing inflammatory processes, endometriosis, lung diseases, and the presence of other malignancies. It should be noted that, both in our research and in that of other scientists, the preoperative HE4 concentrations, compared to CA125, correlate better with the abovementioned prognostic factors of endometrial cancer, which may result in a more individualized treatment of patients [7,8,9,10,11,12,13]. Stiekema et al. found that the sensitivity and specificity of the serum HE4 concentration for differentiation between the superficial and deep invasion of the myometrium were high (61%, 60%), with a negative predictive value of 7% [14]. The sensitivity of ultrasound examination in endometrial cancer was found to be 69%. Therefore, there is a need to use other studies that are more sensitive and will allow proper identification of a higher number of patients. Magnetic resonance imaging is a study that improves sensitivity (71–85%), specificity (72–90%), positive values (51–79%), and negative values (83–85%) in the diagnostic processes of patients with endometrial cancer [15,16,17,18,19,20]. Brennan et al. confirmed these results in a large prospective study by ANECS (Australian National Endometrial Cancer Study), which involved 373 patients with endometrial cancer. They stated that HE4 levels were significantly higher in patients with more advanced endometrial cancer, and that this marker was a better predictor of myometrial infiltration than the CA125 marker [21]. A study by Fanfani et al. [22] evaluated the prognostic value of HE4 and CA125 markers with pathological prognostic factors to complete the preoperative clinical panel and to help with the treatment planning. The researchers have found that a preoperative evaluation of HE4 levels could help stratify patients with deep invasion and/or metastatic disease, and that HE4 concentration was correlated with other relevant prognostic factors that should be considered in choosing an adequate surgical strategy. Caprignole et al. [23] conducted a literature review on the information available on HE4, which confirmed that the use of serum HE4 seems to have good performance in the prognosis and monitoring of patients with endometrial cancer, helping to schedule the appropriate timing of imaging and surgery in a more individualized fashion.

Angioli and Plotti et al. recommended the creation of REM or refined REM B algorithms according to HE4 serum concentration and endometrium thickness, which was intended to stratify patients to low or high endometrial cancer risk groups [24,25]. Angiolli found that HE4 concentration correlated with the depth of myometrial infiltration [26]. Moreover, in two different studies, the researchers found that the concentration of HE4 did not correlate with the involvement of lymph nodes and the FIGO stage of endometrial cancer [27,28]. However, these studies enrolled small groups of patients and small groups with performed lymphadenectomy. In our previous research, which included a smaller group of patients, serum HE4 did not differentiate between patients with G1 and patients with G2 endometrial cancer. In this study, the concentration of HE4 correlated significantly with the grade of endometrial cancer [29]. Similar findings were reported by other investigators. Scientists are constantly debating whether it is worth conducting in-depth complimentary research during patients’ follow-up. Most of them claim that, despite the creation of additional costs by imaging tests (CT, NMR) or routine vaginal smear tests, this does not reduce the cost of treating patients with recurrence of the disease. Of course, the right direction seems to be continuing the search for serum markers that will allow for catching early relapses of asymptomatic patients with a small mass of the recurrent tumor. This would allow for a more effective treatment of patients, with a simultaneous reduction of its cost. Already in 2011, Bignotti et al. suggested that the HE4 concentration, alone or in combination with the concentration of CA125, could predict the prognosis of patients with endometrial cancer (as in patients with lung adenocarcinoma or epithelial ovarian cancer) [30]. Since then, numerous studies have tried to assess whether the preoperative HE4 concentration is an independent prognostic factor in patients with endometrial cancer. In our study, we have shown that HE4 correlated much better with disease recurrence than CA125. Using a multivariate analysis, we found that HE4 concentration measured at each point of the study (median, 70 pmol/L cut-off, 75th percentile, and 95th percentile of baseline values) correlated with the overall survival of patients with endometrial carcinoma. In the case of the CA125 marker, the correlation with DFS and OS was achieved for the median and cut-off point of 35 IU/mL. Two other publications have concluded that HE4 could be an independent prognostic factor for both DFS and OS [21,31]. Brennan et al. showed that an increased level of HE4 was found in 80% of recurrences, while an increase in CA125 marker was found only in 47% of cases [21]. This is slightly different from our results. In our study, at the time of recurrence diagnosis, the increase in HE4 was confirmed in 84% of cases, and of the CA125 marker in 56%. Regardless of the percentage difference, HE4 is a marker that increases much better and faster in the case of relapse. It should be emphasized that the increase of HE4 marker in patients with distant metastases was 100%, and the level of CA125 increased by 71%. In the case of local recurrence, the percentage distribution was slightly different, and equaled 74% for HE4 and 51% for CA125. The limitation of our study was a relatively small percentage of patients with distant metastases. A meta-analysis conducted by Dai et al. revealed that HE4 is a good prognostic factor not only for patients with endometrial cancer, but also those suffering from ovarian, lung, and gastric cancer [31,32].

In summary, the therapeutic problem remains the most difficult for patients that face disease recurrence of endometrial cancer. The results of our study suggest that the preoperative levels of HE4 adjusted for patient characteristics, such as the clinical advancement of the patient, seems to be a good prognostic factor. Additionally, other, already known high-risk factors should be taken into consideration upon deciding on the possible use of adjuvant therapies in endometrial cancer. The determination of HE4 seems to be low-cost, and could help to distinguish the groups of patients in whom treatment should be intensified in order to postpone, or even avoid, the recurrence of the malignancy.

## 5. Conclusions

In patients suffering from endometrial cancer, HE4 is definitely a better prognostic factor than CA125. Lowering HE4 to the levels below the cut-off value of 70 pmol/L and CA125 to <35Iu/mL positively correlated with patients’ DFS and OS. Moreover, the HE4 cut-off value of 186 pmol/L correlated with the possibility of cytoreductive surgery in patients with recurrent endometrial cancer.

## Figures and Tables

**Figure 1 diagnostics-11-00626-f001:**
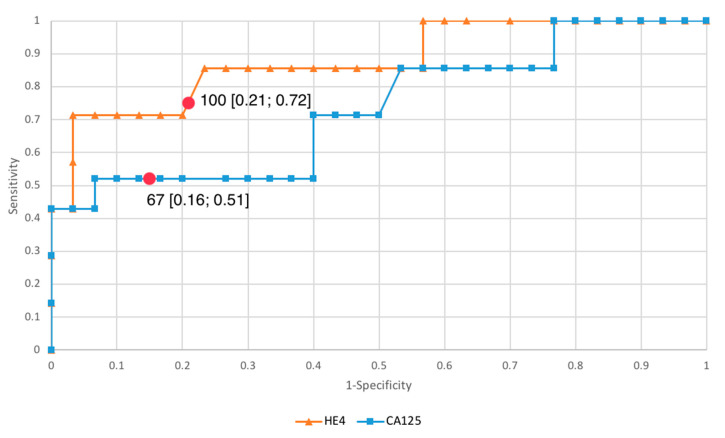
ROC curves for HE4 and CA125.

**Figure 2 diagnostics-11-00626-f002:**
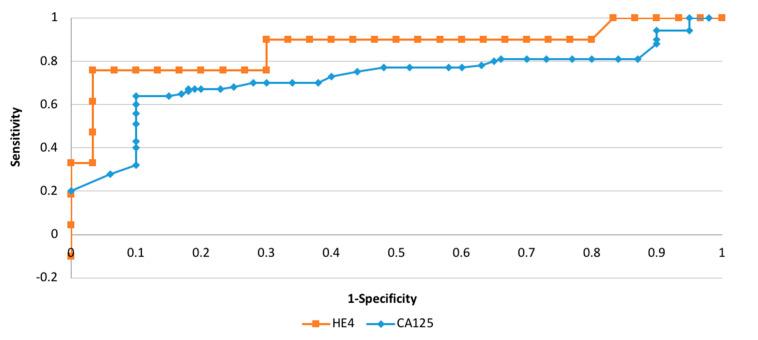
The ROC curves for HE4 markers depending on staging (AUC = 0.88).

**Figure 3 diagnostics-11-00626-f003:**
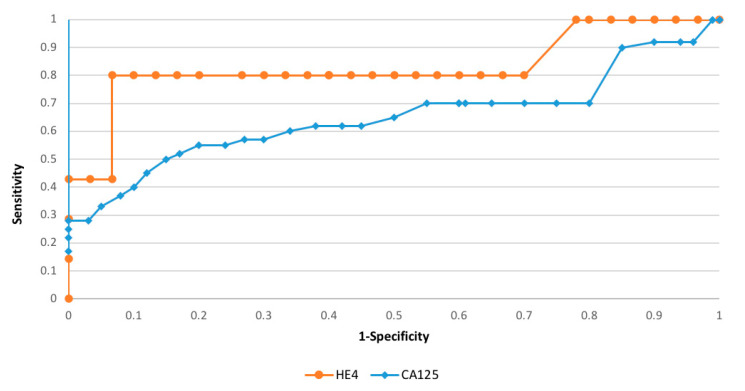
The ROC curves for HE4 markers depending on grading (AUC = 0.78).

**Figure 4 diagnostics-11-00626-f004:**
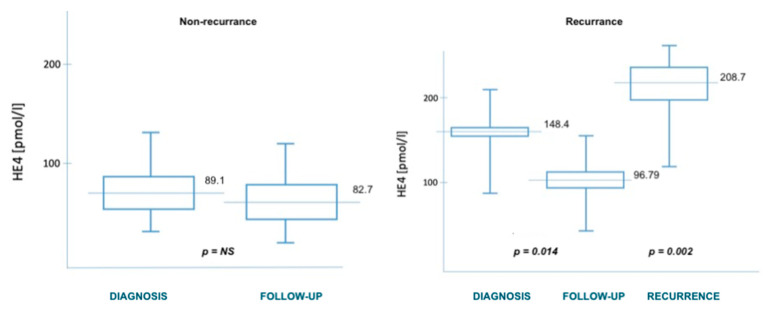
HE4 levels in relation to recurrence.

**Figure 5 diagnostics-11-00626-f005:**
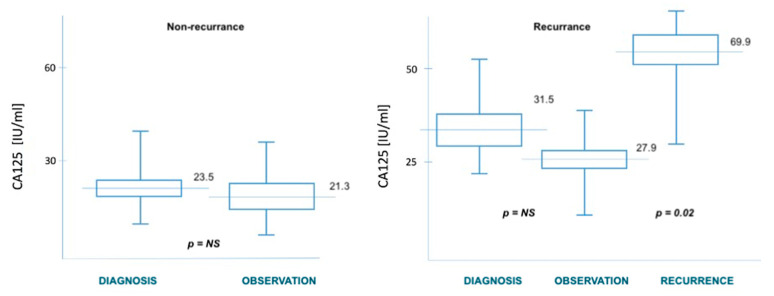
CA125 levels in relation to recurrence.

**Figure 6 diagnostics-11-00626-f006:**
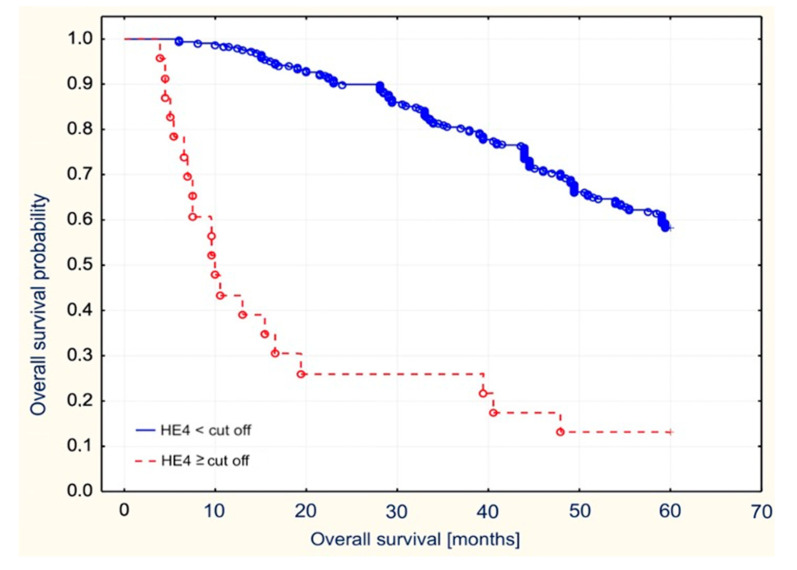
Overall survival stratified by median HE4 values in the examined patients with endometrial cancer.

**Figure 7 diagnostics-11-00626-f007:**
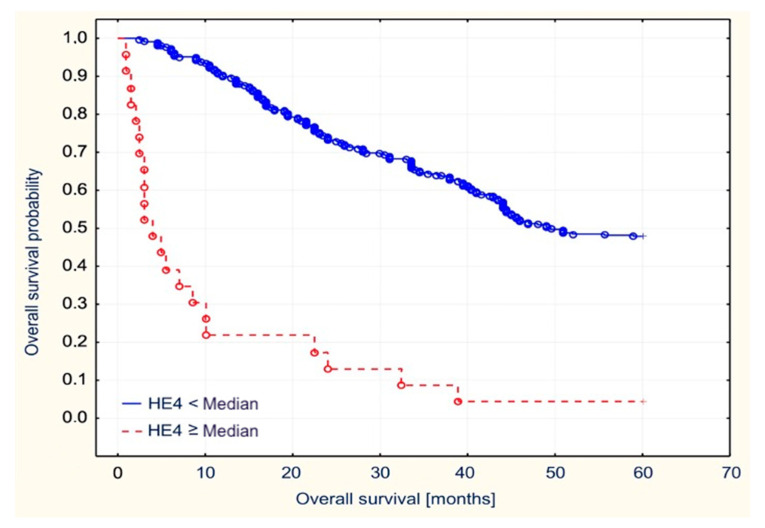
Overall survival stratified by the cut off HE4 value in the examined patients with endometrial cancer.

**Table 1 diagnostics-11-00626-t001:** Clinicopathological characteristics of the patients with endometrial cancer divided into subgroups.

Subgroups	Number of Patients
**Histopathological type**	
Type I (endometrial endometrioid adenocarcinoma)	302
Type II (serous endometrial carcinoma, squamous adenocarcinoma and clear cell carcinoma)	47
**Tumor grading**	
G1	53
G2	225
G3	71
**Clinical staging**	
FIGO I and II	287
FIGO III and IV	62
**Myometrial infiltration**	
Superficial myometrial infiltration (<1/2 of the thickness)	56
Deep myometrial infiltration (>1/2 of the thickness),	293
**Lymph vessel involvement**	
Yes	102
No	247
**Lymph node metastases**	
Yes	62
No	287

**Table 2 diagnostics-11-00626-t002:** Comparative analysis of both markers, HE4 and CA125, as prognostic factors.

	Mean HE4	Median HE4	*p*-Value	Mean CA125	Median CA125	*p*-Value
**Histopathological type**			NS			NS
**Type II**	99.6	99.7		121.2	143.6	
**Type I**	43.8	44.1		48.1	51.6	
**Grading**			0.04			NS
**G2**	64.7	72.1		103.4	99.4	
**G1**	26.7	27.4		38.5	40.0	
**Grading**			0.001			0.02
**G3**	64.7	72.1		166.2	199.1	
**G1**	26.7	27.4		43.8	46.7	
**FIGO staging**			0.001			0.003
**III and IV**	116.4	113.8		289.3	295.3	
**I and II**	34.8	32.3		78.2	72.9	
**Lymph vessels invasion**			0.007			NS
**No**	189.7	167.2		115.1		
**Yes**	61.2	53.5		43.9		
**Lymph nodes metastasis**			0.042			0.01
**No**	208.9	214.5		132.1	141.3	
**Yes**	86.1	88.3		30.9	32.6	
**Myometrium infiltration**			0.001			0.02
**deep**	121.6	118.6		247.2	233.9	
**superficial**	31.1	34.1		67.9	70.9	

NS: not significant.

**Table 3 diagnostics-11-00626-t003:** FIGO stage and recurrence.

FIGO Stage	Patient *n*	Recurrence Rate %	HE4 Median (pmol/L)	CA125 Median (U/mL)
I	184	28.26	109.7	59.9
II	103	55.33	249.6	82.3
III	54	75.90	301.6	92.3
IV	8	87.51	333.8	161.4

**Table 4 diagnostics-11-00626-t004:** HE4 and CA125 serum concentration according to recurrence or nonrecurrence of the cases.

	HE4 Median (Range)			CA125 Median (Range)		
	Recurrence (*n* = 101)	Recurrence- Free (*n* = 186)	*p*-Value	Recurrence (*n* = 101)	Recurrence-Free (*n* = 186)	*p*-Value
diagnosis	92.3 (70.6–101.3)	55.1 (41.8–69.9)	0.002	31.2 (22.9–42.3)	23.5 (18.1–29.9)	NS
post-surgery	74.6 (59.9–92.0)	42.3 (36.2–54.8)	0.001	24.8 (17.2–40.8)	18.9 (11.8–25.6)	NS
observation	102.7 (82.3–118.6)	53.1 (40.8–66.2)	0.001	27.9 (20.6–45.2)	21.3 (14.8–34.5)	0.048
recurrence	267.9 (199.1–289.3)	NA		69.9 (57.9–80.3)	NA	
locoregional recurrence (vaginal, pelvic)	212.9 (187.5–232.4)	NA		63.1 (51.6–77.4)	NA	
distant recurrence (lung, bones)	324.6 (301.7–342.8)	NA		123.4 (102.2–140.3)	NA	

NA: not appropriate.

**Table 5 diagnostics-11-00626-t005:** Cox regression analyses of the disease-free survival and overall survival of classic prognostic factors and HE4.

	Univariate Analysis (Cox Regression Model)					
	DFS			OS		
	HR	95% CI	*p*-Value	HR	95% CI	*p*-Value
**Age**	1.03	0.69–1.06	0.083	1.1	0.98–1.13	0.04
**Stage I, II vs. III, IV**	2.93	1.3–3.58	0.024	2.13	1.68–2.27	0.003
**Grade 1 vs. 3**	1.31	1.1–1.67	0.043	1.61	1.34–2.45	0.01
**Metastases to lymph node**	2.21	1.43–2.98	0.002	2.02	1.57–2.54	0.001
**HE4 median**	1.96	1.52–2.23	0.015	1.83	1.22–2.01	0.004
**HE4 cut off** **(70 pmol/L)**	2.08	1.78–2.44	0.001	1.91	1.63–2.42	0.03
	**Multivariate Analysis (Cox Regression Model)**					
	**DFS**			**OS**		
	**HR**	**95% CI**	***p*-Value**	**HR**	**95% CI**	***p*-Value**
**HE4 median**	1.34	0.8–1.63	0.01	1.21	0.7–1.71	0.003
**HE4 75 percentile**	0.98	0.56–1.04	NS	1.18	0.60–1.45	0.03
**HE4 95 percentile**	1.41	1.12–1.76	0.04	1.62	0.74–1.88	0.022
**HE4 cut off** **(70 pmol/L)**	2.31	1.99–2.76	0.01	1.89	1.08–2.06	0.004

DFS: disease-free survival, OS: overall survival, HR: hazard ratio.

**Table 6 diagnostics-11-00626-t006:** Cox regression analyses of the disease-free survival and overall survival of classic prognostic factors and CA125.

	**Univariate Analysis (Cox Regression Model)**					
	**DFS**			**OS**		
	**HR**	**95% CI**	***p*-Value**	**HR**	**95% CI**	***p*-Value**
**Age**	1.04	0.83–1.23	NS	1.12	0.87–1.71	0.043
**Stage III, IV** **vs. I, II**	2.13	1.51–2.42	0.003	2.06	1.49–2.22	0.014
**Grade 1 vs. 3**	1.41	1.20–1.88	NS	1.37	0.97–1.52	NS
**Metastases to lymph node**	1.92	1.32–2.24	0.045	1.76	1.35–1.90	0.023
**CA125 median**	1.76	1.41–1.99	0.033	1.42	1.19–1.68	0.025
**CA125 cut off** **(35 UI/mL)**	1.82	1.52–2.32	0.002	1.98	1.66–2.31	0.018
	**Multivariate Analysis (Cox Regression Model)**					
	**DFS**			**OS**		
	**HR**	**95% CI**	***p*-Value**	**HR**	**95% CI**	***p*-Value**
**CA125 median**	1.11	0.89–1.33	NS	1.22	0.97–1.49	0.024
**CA125** **75 percentile**	0.98	0.77–1.24	NS	1.01	0.68–1.20	NS
**CA125** **95 percentile**	1.04	0.82–1.30	0.012	1.13	0.79–1.34	NS
**CA125 cut off** **(35 UI/mL)**	1.45	1.20–1.63	0.026	1.38	0.81–1.55	0.037

## Data Availability

The data presented in this study are available on request from the corresponding author.

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
