# Peer review of "Can the Determination of HE4 and CA125 Markers Affect the Treatment of Patients with Endometrial Cancer?"

_diagnostics, 2021, doi:10.3390/diagnostics11040626_

Round 1

Reviewer 1 Report

I carefully read and evaluated the paper “Can the determination of HE4 and CA125 markers affect the treatment of patients with endometrial cancer?”. The principal aim of this paper is to evaluate the use of HE4 and CA125 as prognostic factors in patients with endometrial cancer at different stages. Similar studies already exist in the literature. I have the following considerations:

  • In the “introduction” section the authors affirmed: “These markers are commonly used in patients’ monitoring during the chemotherapeutic treatment for ovarian cancer”. In international guidelines, the use of HE4 is not mandatory. The authors should specify this more accurately.
  • In the "Patient characteristics" section they only talk about consents while the actual characteristics of the patients should be specified.
  • The inclusion and exclusion factors should be specified more clearly.
  • The number of patients enrolled with advanced endometrial cancer is not very consistent.
  • In Figure 2 and Figure 3 the subdivisions into staging and grading are not very clear.
  • Figure 6 and figure 7 are overlapped, so they cannot be evaluated.
  • What kind of medical / surgical treatment have the patients been subjected to?
  • In the discussion: why were patients with endometriosis or other conditions that alter CA125 values not excluded from the study?
  • “We believe… imaging studies”: I believe that the data reported in this study are not sufficient to support this sentence.
  • Some references should be updated, and implemented. Such as the work of Fanfani et al; or Capriglione et al. etc
  • I suggest reviewing English language.

Author Response

Dear reviewer, we would like to thank you for your comments

  • In the “introduction” section the authors affirmed: “These markers are commonly used in patients’ monitoring during the chemotherapeutic treatment for ovarian cancer”. In international guidelines, the use of HE4 is not mandatory. The authors should specify this more accurately.

We deleted this information and added the following: 

The above-mentioned markers are commonly used to differentiate malignant adnexal lesions in clinical practice. They can also be used to monitor neo- and adjuvant treatment of ovarian cancer patients.

  • In the "Patient characteristics" section they only talk about consents while the actual characteristics of the patients should be specified.
  • The inclusion and exclusion factors should be specified more clearly.
  • The number of patients enrolled with advanced endometrial cancer is not very consistent.

Thank you for your comments; we improved patient characteristics section and added the following information:

All of the patients enrolled in the study underwent endometrial biopsy, abrasions or hysteroscopy. Based on the obtained histopathological results, they were qualified for radical surgery involving removal of the uterus with appendages and hip salping in less advanced stages. In advanced stages of endometrial cancer, radical hysterectomy and lympadenectomy were performed. Total pelvic exenteration was performed in stage IV patients. Patient analysis was conducted depending on the hormonal status, dividing the population into two subgroups: pre- and postmenopausal patients. Patients were qualified as postmenopausal if their last menstrual period was latter than 12 months ago or if their FSH level was above 30U / L. Patients aged 36 to 79 were enrolled in the study. The mean age of patients enrolled in the study was 60.8 years.

Please see the reviewed version of the manuscript

  • In Figure 2 and Figure 3 the subdivisions into staging and grading are not very clear. 

We deleted some of the informations to make the tables more visible and accessible

  • Figure 6 and figure 7 are overlapped, so they cannot be evaluated.

We changed the positions of the figures.

  • What kind of medical / surgical treatment have the patients been subjected to?

We added this information in patient characteristics

  • In the discussion: why were patients with endometriosis or other conditions that alter CA125 values not excluded from the study?
  • We added the following information:

Moreover, one patient was excluded due to rheumatoid arthritis , one due to miastenia gravis and 4 patients due to endometriosis.  One patient was also excluded due to idiopathic pulmunary fibrosis. Ultimately, 349 patients were enrolled in the study. Table 1 presents the detailed characteristics of the patients.

  • “We believe… imaging studies”: I believe that the data reported in this study are not sufficient to support this sentence.

we deleted this sentence from the manuscript

  • Some references should be updated, and implemented. Such as the work of Fanfani et al; or Capriglione et al. etc

We added the suggested bibliography

  • I suggest reviewing English language.

We further edited the language used in the manuscript. 

Reviewer 2 Report

Good paper omn an interesting topic. The only bias is that it is not clearly stated the selection criteria for lynohadenectomy

Author Response

Dear reviewer,

Thank you for your comment

We added some additional information to the manuscript, please see the improved version

Round 2

Reviewer 1 Report

The authors have made the requested changes